# In Vivo Pharmacodynamics of *Calophyllum soulattri* as Antiobesity with In Silico Molecular Docking and ADME/Pharmacokinetic Prediction Studies

**DOI:** 10.3390/ph16020191

**Published:** 2023-01-28

**Authors:** Inarah Fajriaty, Hariyanto Ih, Irda Fidrianny, Neng Fisheri Kurniati, Muhammad Andre Reynaldi, I Ketut Adnyana, Rommy Rommy, Fransiska Kurniawan, Daryono Hadi Tjahjono

**Affiliations:** 1Department of Pharmacology and Clinical Pharmacy, School of Pharmacy, Bandung Institute of Technology, Jl. Ganesha 10, Bandung 40132, Indonesia; 2Department of Pharmacy, Faculty of Medicine, Universitas Tanjungpura, Pontianak 78124, Indonesia; 3Department of Pharmaceutical Biology, School of Pharmacy, Bandung Institute of Technology, Jl. Ganesha 10, Bandung 40132, Indonesia; 4Department of Pharmacochemistry, School of Pharmacy, Bandung Institute of Technology, Bandung 40132, Indonesia

**Keywords:** *Calophyllum soulattri*, antiobesity, in vivo, in silico

## Abstract

This study aims to determine the antiobesity activity of *Calophyllum soulattri* leaves extract (CSLE) on high fat diet-fed rats (HFD) and to predict the molecular docking and pharmacokinetics of selected compounds of *Calophyllum soulattri* to fat mass and obesity-associated protein (FTO). Daily body weight, organ, carcass fat (renal and anal), body mass index, total cholesterol, and total triglyceride levels were observed after CSLE was given orally for 50 days. Furthermore, body mass index of a CSLE dose of 50 mg/kgbw, 100 mg/kgbw and orlistat (120 mg/kgbw) group are 0.68, 0.57 and 0.52, respectively. The total body weight of the CLSE dose of 100 mg/kgbw group showed the lowest percentage change, followed by a CLSE dose of 50 mg/kgbw compared to the normal and positive control group. The carcass fat index of CSLE dose of 100 mg/kgbw was not significantly different from orlistat, which was in line with its total cholesterol level and triglyceride (*p* < 0.05). The binding affinity of selected compounds from *Calophyllum soulattri* (friedelin, caloxanthone B, macluraxanthone, stigmasterol, trapezifolixanthone, dombakinaxanthone, and brasixanthone B) to FTO are –8.27, –9.74, –8.48, –9.34, –8.85, –8.68 and –9.39 kcal/mol, which are better than that of orlistat at –4.80 kcal/mol. The molecular dynamics simulation showed that the interaction between Caloxanthone B compounds and obesity receptors was relatively stable. Lipinski’s rule determined the absorption percentage of all compounds above 90% with good drug-likeness. The results showed the potential of CSLE as an antiobesity drug candidate.

## 1. Introduction

The prevalence of obesity and metabolic disorders has increased worldwide, associated with an unhealthy lifestyle, such as excessive fast food consumption and low level of physical activities [1,2]. The World Health Organization (WHO) suggests that the prevalence is approximately 13% of the world’s adult population [3]. Obesity is a major risk factor for many chronic diseases, including cardiovascular diseases, which are the leading cause of death worldwide [4]. Drugs and chemical treatments have been used to overcome these health problems. The development of antiobesity agents has been conducted for over a decade despite its side effects, which include diarrhea, steatorrhea, abdominal cramps, fat-soluble vitamin deficiencies, fecal incontinence, and flatulence [5]. However, these adverse gastrointestinal side effects are limiting and tolerable to many patients [6]. Along with the research on antiobesity chemical-based compounds, new alternative drugs based on plants were also established [7,8].

*Calophyllum soulattri* is an endemic plant in Indonesia, and its leaves extract is commonly used for the treatment of obesity containing several secondary metabolite compounds, including flavonoid, steroid, triterpenoid, phenol, tannin, and saponin [9] Friedelin, a pentacyclic terpenoid constituent, has been found in *Calophyllum soulattri* leaves extract, where this compound was assumedly involved in the hypolipidemic activity of *Azima tetracantha* Lam. In rodent hyperlipidemia experimental models [10,11] *Calophyllum soulattri* showed a good effect as an antihyperlipidemic candidate through its secondary metabolites, confirmed by the recent in vivo study on male Wistar rats [12]. Moreover, several compounds of *Calophyllum soulattri* with good antioxidant ability, including friedelin, caloxanthone B, macluraxanthone, stigmasterol, trapezifolixanthone, lambkinaxanthone, and brasixanthone B, are known to exhibit potential antiobesity properties [13,14,15]. Acute and subchronic toxicity studies of *Calophyllum soulattri* leaves extract revealed no mortality or observed clinical signs of toxicity during investigation periods with LD50 > 5000 mg/kgbw, but hepar necrosis was associated with the long-term use of high-dose CSLE, such as 1000 mg/kgbw transaminase. Moreover, the reported dose of *Calophyllum soulattri* leaves extract was between 100 and 400 mg/kgbw with no adverse effects appearing from long-term use [12]. There are no reported studies describing the properties of *Calophyllum soulattri* as antiobesity agents. Therefore, this study aimed to use an in vivo and in silico combined approach to investigate the potential antiobesity activity of *Calophyllum soulattri.* The antiobesity effects of the ethanolic extract leaves was evaluated on high-fat diet-fed rats (HFD). The interactions between selected potential compounds and obesity-associated protein (FTO) as the target receptor for antiobesity drugs were predicted by molecular docking simulations. Furthermore, in silico absorption, distribution, metabolism, and excretion (ADME), toxicity and drug-likeness predictions were also determined.

## 2. Results

The specific and non-specific parameters results of the 96% ethanolic extract of *Calophyllum soulattri* leaves (CSLE) are shown in Table 1. We performed the organoleptic determination of CSLE and its solubility towards two different organic solvents as the specific parameter of the extract. The organoleptic character describes colour, odour and texture, while the solubility parameters are used to determine the availability of polar and semi-polar (or non-polar) compounds in the final extract. The results showed a similar percentage value of the CSLE contents that were soluble in water and ethanol, which were 23.1% and 24.0%, respectively. Furthermore, using polar and semipolar solvents will result in a favourable extraction process. 

The non-specific parameters of CSLE, including density extract, drying losses, and water content, have been performed as well, with values of 0.8033 g/mL, 16.32 ± 0.66% and 12.76, respectively (Table 1). The Indonesian Herbal Pharmacopoeia specifies a minimum water content of 10% for extract, showing that the CSLE used in this study satisfies the requirements for standard quality [16].

A review of the constituent compounds of CSLE was intended through phytochemical screening. According to phytochemical screening results, the 96% ethanolic extract of *Calophyllum soulattri* leaves contained alkaloids, polyphenols, flavonoids, steroid-triterpenoids, and saponins (Table 2).

Mechanically agitated foam often results in an unsteady thermodynamic system. The foam will degrade while it is inactive. Furthermore, the stability of the foam is determined by its thinning rate [17]. Table 3 displays the height change of the foam over time. CSLE’s foaming index and fish index are higher compared to those of powdered CSL as a result. 

The drug-likeness parameter is an analysis to predict the physico-chemical properties of a drug based on its molecular weight (MW), Log P, the number of hydrogen donors (H donors) and acceptors (H acceptors). Based on these parameters, as shown in Table 4, all seven compounds in *Calophyllum soulattri* and orlistat complied with Lipinski’s rule. 

The molecular docking simulation results were determined by the Gibbs free energy value (ΔG), where the compound with the lowest ΔG value was predicted to have the strongest and most stable binding to the receptor. Molecular docking results showed that the seven compounds in *Calophyllum soulattri* have a better binding affinity than orlistat as a positive control, with the lowest ΔG shown by caloxanthone B (Table 5). Moreover, this energy affinity describes the bond energy that occurs between the compounds in *Calophyllum soulattri* and the FTO receptor.

Table 6 explained the pharmacokinetics of selected compounds of *Calophyllum soulattri*, including their water solubility, intestinal absorption, volume of distribution, total clearance, and the prediction of their toxicity, including general toxicity (LD50) and specific toxicity (AMES toxicity test and hepatotoxicity). Overall, an intestinal absorption value above 90% was shown by all selected compounds. Based on the AMES toxicity results, four compounds are predicted to be mutagens, such as caloxanthone B, macluraxanthone, dombakinaxanthone, and brasixanthone B, but further in vivo tests are required. However, the prediction of hepatotoxicity test showed that most of the compounds are not toxic, except for dombakinaxanthone. The prediction of pharmacokinetic and toxicity parameters was performed based on the similarity of the chemical structure of the predicted compounds to the chemical structure contained in the pkCSM-Biosig Lab. These predictions provide a basis for analysing pharmacokinetic and toxicity parameters based on seven potential compounds in *Calophyllum soulattri.*

## 3. Discussion

### 3.1. Calophyllum Soulattri Leaves Extract (CLSE) as Potential Antiobesity Agent

The standardization of specific and non-specific parameters of *Calophyllum soulattri* leaves extract (CLSE) was performed to fulfil the requirements as a herbal raw material [16]. It was conducted at three different harvesting times, including the content of dissolved compounds and the chemical content of extracts as specific parameters, while the non-specific parameters included water content, density and drying losses, as summarized in Table 1. 

Chromatogram patterns were also standardized as specific parameters. In contrast, chemical content can be detected under UV wavelengths of 254 nm, but only under UV wavelengths of 366 nm, as shown in Figure 1.

The foaming index was demonstrated to determine the capability of an extract to produce foam. In contrast, the fish index was obtained by the ability of the most diluted extract to kill three of five *Tilapia mossambica* in one hour [18]. The foaming index results showed the presence of saponin (Table 2). However, our results showed no toxic activity since their fish index value was lower than 1000 (Table 2).

A high-fat diet (HFD) induction was given to induce the body weight of rats as the obese animal model used and to increase the total blood cholesterol and triglyceride value. BMI measurement was calculated on the same day before termination. Results for BMI can be seen in Figure 2.

There is no significant difference in BMI value for all groups on day 0. There were no significant differences shown between the positive control (Orlistat) and CSLE dose 100 mg/kgbw on day 50 (* *p* < 0.05). The results demonstrated that BMI for all groups on day 0 was not significantly different (*p* < 0.05) and was classified as normal with a BMI range of 0.49 to 0.50. However, a BMI increase was observed in all groups on day 50, after 50 days of HFD induction and 20 days of treatment. The BMI of the orlistat group has the lowest value (0.52), while the negative control given HFD induction for only 50 days without any treatment has the highest value (Figure 2). Another promising finding was the CSLE doses of 50 mg/kgbw and 100 mg/kgbw at BMI values of 0.68 and 0.57, respectively. These values were not significantly different compared to a positive and normal control group (*p* < 0.05). Therefore, the doses of CSLE can inhibit the increase of BMI in animals with HFD induction. The body weights of rats were observed each day during the HFD induction period from day 0 to 50 (Figure 3).

Figure 3 illustrates the increasing animal body weight profile after 50 days of HFD induction, except for the negative control. Weight loss in orlistat and both CSLE groups were observed compared to normal control during the test period. Furthermore, the percentage of weight change from day 0 to 50 was calculated, as seen in Figure 4.

The normal control group showed up to 33.98% of initial body weight, whereas the negative control increased to 54%. The orlistat group had an increased percentage of body weight of up to 27.77%, but in line with the BMI results, CSLE doses of 50 mg/kgbw and 100 mg/kgbw showed a lower percentage change in body weight, at 28.59% and 24%, respectively. The result indicates that CSLE doses can inhibit the weight increase induced by HFD. However, these values were not significantly different compared to the positive control groups (* *p* < 0.05). This suggests that both doses of CSLE can inhibit the increase of body weight in animals with HFD induction.

Obesity is a condition where there is an increase in adipose tissue mass [19]. Accumulating fat or triacylglycerol or triglycerides is the cause of excessive weight gain because other food reserves, such as glycogen and protein, do not have the potential to accumulate in adipose tissue. In addition, adipose tissue releases adipokines to the body in order to store triacylglycerol that may cause hyperplasia and hypertrophy with different potentials between individuals [20,21]. Gastric and lingual lipase is responsible for the triacylglycerol metabolism of food into free fatty acids and diacylglycerol. Partial digestion in the stomach forms large fat molecules emulsified with bile salts, and the fat develops into small granules. In emulsions, triglycerides and diglycerides of food will form a mixture of polar lipids, phospholipids, cholesterol and free fatty acids encapsulated by oligosaccharides and denatured proteins and bile salts to produce complex molecular compounds. Pancreatic lipase will interact with the hydrolyzed molecule to produce free fatty acids, mono acylglycerol, and diacylglycerol, which binds to cholesterol, bile salts fat-soluble vitamins, and lysophosphatidic acid [21,22]. These fat digestion activities generate fat to enter and accumulate in the body.

Orlistat is a lipstatin derivative that binds covalently to the active side in pancreatic lipase and forms a stable complex [23]. The complex causes a conformational change in the enzyme lipase to be inactive and therefore cannot hydrolyze fats into fatty acids and monoglycerides and is excreted with the feces [24]. This inhibition causes a decrease in fat absorption by 30% and leads to steatorrhea or oily feces. Rats given orlistat also suffer from the same adverse effect. However, the rats administered with CSLE have shown significant weight loss and lower BMI values compared to the orlistat group. Our results demonstrated that CSLE can potentially be developed into an antiobesity agent.

The organ index is one of the parameters observed to determine the ability of a substance to cause adverse effects. The organ index can also indicate the test compound effect, wherein a significant difference between the control and organ index test groups can be different but morphologically not different, as seen in the diagram below.

Figure 5 shows the average organ index of rats in each group. An increase in liver organ index in the orlistat group was observed after the treatment. The data also show an increase in the pancreas index of CSLE at doses of 50 mg/kgbw and 100 mg/kgbw, which is higher than for the normal group. The index provides a general description of the effect of the compound on several organs. It cannot be used as an assay of the damage or the improvement of organ function and should be followed by a histopathology assessment.

In this study, the carcass fat index value was also determined to provide the potential effect of CLSE as an antiobesity agent. The carcass fat index is a parameter observed in obese rats to determine fat deposits, where anal fat is the deposit around rats’ anus, and renal fat is deposited near the kidney.

As an antiobesity agent, orlistat plays a role in reducing fat absorption, which can minimize animal fat storage better than the normal group, as shown in Figure 6. The CSLE dose of 50 mg/kgbw showed decreased fat storage compared to the negative group. However, the CSLE dose of 100 mg/kgbw showed a decrease in renal and anal fat deposits (Figure 6). These values were not significantly different compared to the positive control groups (* *p* < 0.05). The results confirm that a CSLE dose of 100 mg/kgbw can decrease renal and anal fat deposits.

The total cholesterol and triglyceride levels were observed before the drug treatment (day 30) compared to the last day as can be shown in Figure 7 and Figure 8, respectively. The total cholesterol and triglyceride levels of the orlistat group and the CSLE dose of 50 mg/kgbw and 100 mg/kgbw were decreased on day 50. These values were not significantly different compared to the positive control groups (* *p* < 0.05). This is probably because of the lipase inhibition activity of orlistat and CSLE as enzymes involved in lipid hydrolysis, resulting in decreased total cholesterol and triglyceride levels. The results show the potential effect of CSLE in lowering total cholesterol and triglyceride levels compared to orlistat as a positive control. 

Cholesterol and triglycerides are lipids that circulate in large amounts in the blood that are bound to phospholipids and lipoproteins that dissolve in water and can be carried throughout the body. Increased lipids in the blood may cause hyperlipidemia and abdominal fat accumulation [25]. High levels of total cholesterol and LDL (low-density lipoprotein) blood are caused by the overconsumption of saturated fat in the diet. In contrast, the increase in blood triglyceride levels or hypertriglyceridemia is influenced by genes and the consumption of foods, such as carbohydrates, fats, and alcohol [26]. The normal total cholesterol and triglyceride levels in mice’s blood ranged from 50–100 mg/dL [27].

### 3.2. Molecular Docking and Pharmacokinetic Prediction of Selected Potential Compounds of Calophyllum Soulattri

This section will illustrate the molecular docking of selected potential compounds of *Calophyllum soulattri*. These include friedelin, caloxanthone B, macluraxanthone, stigmasterol, trapezifolixanthone, dombakinaxanthone and brasixanthone B to fat mass and obesity-associated protein (FTO), with their pharmacokinetic prediction. These seven compounds were selected based on a previous study related to the phytochemical determination of *Calophyllum soulattri* that has been established and was provided in the Pubchem data bank [13,14,15,28,29].

Re-docking was also performed to validate the docking protocol. It showed that the RMSD value of native ligands on the crystal structure of FTO (PDB ID: 3LFM) was 1.62, implying that the specified docking parameters can be used, since the value is lower than 2 [30]. Furthermore, other parameters of drug-likeness predictions of selected potential compounds with orlistat as a comparative drug can be seen in Table 3.

Based on these data, the selected compounds contained in the star parameters were similar to those in the positive control, where molecular weight (<500), hydrogen donor (<5), and hydrogen acceptor (<10) complied with Lipinski’s rule. Therefore, the selected potential compounds from *Calophyllum soulattri* are predicted to have drug-likeness. The docking results of these compounds chosen to fat mass and obesity-associated protein (FTO) using Autodock 4.2 are exhibited in Table 4.

Based on the docking results, the binding affinity of all selected potential compounds of *Calophyllum soulattri* was lower than that of native ligand and Orlistat. Therefore, it is predicted that these selected compounds have a stronger binding interaction to FTO compared to those of native ligand and orlistat. Caloxanthone B shows the best binding affinity and inhibition constant based on the molecular docking simulation. The binding affinity value explains the strong interaction of these compounds with the receptor [31]. The results of the docking visualization using Discovery Studio 2021 on the best-selected compounds from *Calophyllum soulattri* compared to positive controls from Orlistat is illustrated in Figure 9.

The molecular dynamic was performed during 100 ns of time simulations with the addition of various parameters adjusted to actual body conditions, such as temperature set to 310 K and the presence of solvent. Furthermore, pharmacokinetic and toxicity parameters based on in silico predictions for each selected compound can be seen in Table 5. All of these selected compounds had an adsorption percentage above 90%, hence good absorption is equal to or more than 30% [32]. However, the toxicity prediction showed that several compounds were predicted to be mutagens based on AMES results, and one compound potentially induced hepatotoxicity (Table 5). Therefore, further in vitro and in vivo assays are needed [33,34].

Delta G (**△*G***) was determined from the total of all energy components (Table 7). The energy component of Van der Waals was contributed to from molecular mechanics, while the electrostatic contribution to the solvation free energy determined by Poisson-Boltzmann that calculated by an empirical model from nonpolar contribution. Based on the results of molecular dynamics, the binding free energy of orlistat is more negative than that of the caloxanthone B compounds. The nonpolar group in orlistat also has a higher contribution than the nonpolar group in caloxanthone B. This can be caused by the chemical structure of orlistat, which tends to be more lipophilic than caloxanthone B. However, the interaction of caloxanthone B with the receptor for 100 ns of time simulation still remained in the binding pocket (Figure 10). Important amino acids at the FTO receptor are Tyr 108, Glu 234 and Ser 229. Caloxanthone B and orlistat bind to these amino acids. The bond formed to these amino acids allows for further reactions to occur, which in turn allows for an anti-obesity effect. The free energy of caloxanthone B and orlistat are −22.53 and −34.21 kkal/mol, respectively. This indicates that caloxanthone B is predicted to have a relatively stable binding interaction with obesity receptors [35]. 

## 4. Materials and Methods

### 4.1. In Vivo Study

#### 4.1.1. Extraction of CSL Powder 

*Calophyllum soulattri* leaves (CSL) were collected from the Mandor region, West Kalimantan Province, Indonesia, and authenticated in the Biology Laboratory of the Mathematics and Science Faculty of Universitas Tanjungpura. CSL was washed to clean any materials left in the leaves and was dried for 2 days. Dry sortation was performed to remove unwanted materials. Subsequently, the leaves were powdered using a blender and stored in a dry and well-ventilated area away from direct sunlight. CSL was extracted by continuous extraction using a Soxhlet apparatus with 96% ethanol. Ethanol was used since it is considered as a universal solvent that may attract the potential compounds of the plants evaluated, and this solvent is appropriate for continuous extraction using the Soxhlet apparatus as well. Compared to other solvents, 96% ethanol is safer to use. After exhaustive extraction, the collected extract was dried under reduced pressure using a rotavapor and dried in a water bath [36]. 

#### 4.1.2. Phytochemical Screening and Thin Layer Chromatography (TLC)

Phytochemical screening was performed to detect the presence of alkaloids, steroids, triterpenoids, flavonoids, tannins, quinones, and saponins. The extract was screened by the TLC method using a selected mobile phase n-hexane and ethyl acetate combination at the ratio of 7:3, with the observation performed under λ_254_ nm and λ_366_ nm of UV light. Organoleptic, yield percentage, water and ethanol solubility, loss on drying and density of extracts were also determined as non-specific parameters [37].

#### 4.1.3. Foam Index Test

The foam forming capability of powdered CSL is measured in terms of the index. One gram of powdered CSL was diluted in 100 mL of water, heated for 30 min, and then filtered. Furthermore, 10 tubes were prepared, and 1, 2, 3, 4, 5, 6, 7, 8, 9, and 10 mL of diluted suspension were put into the test tubes, after which 10 mL of water was added into each tube. The test tubes were shacked for 15 s with two strokes per second and were observed for 15 min. The foam index is considered as <100 when the foam layer is <1 cm in the test tube. In addition, the index is considered >1000 when the foam layer is ≥1 cm in the test tube; therefore, further dilution was required. The foaming index is calculated as follows when the foam is more than or equal to 1 cm in the test tube in the most diluted solution [17]
(1)$$\mathrm{Foaming}\;\mathrm{index}=\frac{1000}{a},$$
where a is the volume of extract in the tube (mL)

#### 4.1.4. Fish index Test

A stock solution of 2% was prepared by diluting 4 g of powdered CSL into 200 mL of water and was heated for 30 min. A solution of 1%, 0.5% 0.25%, 0.1% and 0.05% were prepared by adding 50, 25, 12.5, 5, 2.5 mL of stock solution into 100 mL of water. The fish index value was measured by determining the most diluted solvents that kill three of five fish in an hour. *Tilapia mossambica* with a 2–4 cm length was used as a fish test subject [9,14,38].

#### 4.1.5. Antiobesity Test

Rats were acclimatized for 7 days, and the healthy ones without any physical defect were included as animal subjects. Rats were grouped into 5 groups consisting of normal control (chow and water only), positive control (120 mg/kgbw of orlistat), negative control (Carboxymethylcellulose Sodium or CMC Na 1%), CSLE dose 1 (50 mg/kgbw of CSLE), and CSLE dose 2 (100 mg/kgbw of CSLE). Chows of B2 (551) with quail eggs and cattle fat were orally used as a high-fat diet (HFD) induction. Furthermore, the induction with HFD started on day 1, except for rats plotted into normal groups. The CSLE and Orlistat were administered orally. Treatments were given after 30 days of chow, water and HFD feeding and continued for 50 days before termination. 

Rats were fed 20 g of chow daily with water supply ad libitum for 50 days. However, CSLE was diluted in CMC Na 1% and was administered orally to the animal model for 20 days at 50 mg/kgbw and 100 mg/kgbw. Orlistat, as the comparative drug, was given at a dose of 120 mg/kgbw for 20 days. Parameters observed included rat body mass index, body weight, organ index, carcass fat index, blood triglyceride and total cholesterol value. All animal subjects were sacrificed after 50 days of experiments. Fat around the kidney and the lower area was measured for perirenal and perianal fat index determination. The pancreas, kidney, spleen, liver, heart and lungs were measured for organ index determination. Colourimetric data were used for the determination of blood triglyceride and total cholesterol value. Data are expressed as mean ± standard deviation (SD) for each group and statistically analyzed with a one-way analysis of variance (ANOVA). All analyses and comparisons were evaluated at a *p*-value of 5% (*p* < 0.05). The protocol was peer-reviewed by the Animal Ethics Committee of the Faculty of Medicine of Universitas Tanjungpura with an ethical clearance number of 588/UN22.9/TA/2021 [39].

### 4.2. In Silico Study

#### 4.2.1. Molecular Docking

Molecular docking of the selected potential compounds of *Calophyllum soulattri*, such as friedelin, caloxanthone B, macluraxanthone, stigmasterol, trapezifolixanthone, dombakinaxanthone and brasixanthone B, with orlistat as a comparative drug, to fat mass and obesity-associated protein (FTO) were analyzed and visualized using Autodock 4.2 software and Discovery Studio 2021. DOI for the FTO PDB DOI: 10.2210/pdb3LFM/pdb accessed on 12 November 2021. The ligands were accessed in https://pubchem.ncbi.nlm.nih.gov/ on 3 November 2021. The three-dimensional structure of the target FTO receptor (PDB ID: 3lfm) was obtained from the RSCB protein data bank. Selected compounds were obtained from the PubChem website to obtain a three-dimensional structure in pdb format. Active site detection, docking and analysis of the bound receptor amino acids was carried out using the Autodock computer program (Version 4.2, updated for version 4.2.6) with the help of AutoDockTools version 1.5.6 performed on the potential active site (cavity) detected at the FTO receptor via ligand natively bound in the receptor structure. Assessed energy involved via Gibbs free energy (kcal/mol). The initial stage for molecular docking is redocking the native ligand before docking the compound to be tested. Redocking was performed to validate that the selected docking parameters are appropriate and can applied for the compound to be tested. The criteria for the overlapping RMSD value of redocking results that meet the standard are less than 2 A [40]. The docking simulation was carried out with the 100 conformation searches, meanwhile, 100 compound conformations were used in the molecular docking, and the best conformation was selected for interaction analysis and a binding affinity study. The grid box spacing parameter was 0.375 A, the x-axis: 29.043; y-axis: −6.644; and z axis: −29.329, and the maximum number of medium-level energy evaluations (2,500,000). The algorithm used was the Lamarckian genetic algorithm. The analysis and visualization of the docking results used the BIOVIA Discovery studio 2021 program [41].

#### 4.2.2. ADMET Prediction

The parameter analysis of each compound, including absorption, distribution, metabolism, excretion, and toxicity were predicted with the pkcsm. Molecules input data for prediction through pkcsm with smiles format was then entered and analyzed on the website https://biosig.lab.uq.edu.au/pkcsm/prediction accessed on 15 January 2022 [42].

#### 4.2.3. Molecular Dynamics Simulation

The best molecular docking compounds were then further analyzed by molecular dynamics to determine the stability of the compound binding to the FTO receptor by providing external factors that exist in the body using Amber16 software. Factors that are regulated on the molecular dynamics include pH adjusted to body pH, which is 7.4, the body temperature is 37 °C, the addition of solvent and ions in the body and the duration of the interaction is set to 100 ns simulation. The initial stage for dynamic molecular simulation is the preparation of ligands and receptors. The FTO receptor is formatted according to the Amber16 program. The best compound from docking and positive control of orlistat was optimized by geometry using the method contained in the Amber16 program. The results of the data from the analyzed molecular dynamics are the Gibbs free energy obtained using the MMPBSA method and the RMSD graph. Parameterization used the AMBERff14SB force field for proteins, while the ligand parameterization was carried out using the AMBER force field (GAFF). The preparation stage includes the minimization stage, heating to a temperature of 310 K, temperature equilibration, pressure equilibration, and continuation with the simulation process. The simulation process (production run) was carried out with a timestep of 2 fs for 100 ns. In the dynamic molecular simulation, the water model used is TIP3P and the system is neutralized by adding Na^+^ ions for neutralization if the charge is negative and Cl^–^ ions for neutralization if the charge is positive. The results of the molecular dynamic simulation were then visualized and analyzed using UCSF Chimera and BIOVIA Visualizer 2021. An RMSD analysis was carried out to observe the stability of the ligand-receptor interaction over time. The calculation of the value of Gibbs free energy was carried out using the Molecular Mechanis Poisonn-Boltzmann Surface Area (MM-PBSA) method. Calculations were performed on 1000 frames in the last 10 ns [43,44,45].

## 5. Conclusions

The metabolite compounds in CSLE were alkaloids, flavonoids, polyphenol, terpenoids, and saponins. CSLE may inhibit the increasing BMI, body weight percentage, anal carcass fat, and total blood cholesterol with the best activity shown by the dose of 100 mg/kgbw but does not affect the triglyceride levels. The molecular docking and molecular dynamic simulations of the selected potential compounds of *Calophyllum soulattri* binding to fat mass and obesity-associated protein (FTO) showed that caloxanthone B has the best affinity compared to those of native ligand and Orlistat. These selected compounds also have a drug-likeness compared to orlistat based on Lipinski’s rule, with the prediction of absorption in the intestinal tract being more than 90%.

## Figures and Tables

**Figure 1 pharmaceuticals-16-00191-f001:**
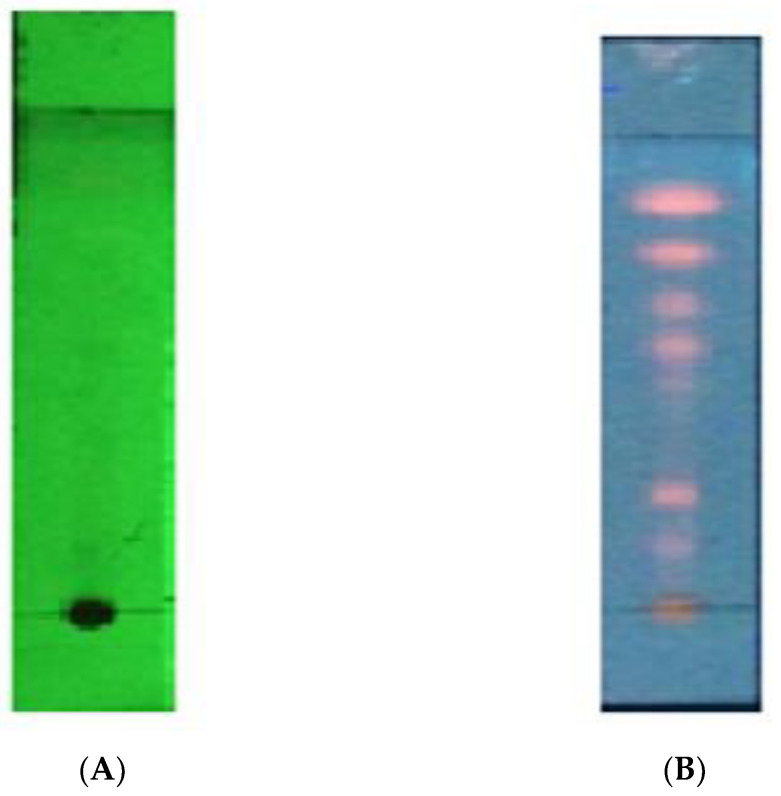
TLC chromatogram pattern of CLSE under UV wavelengths of 254 nm (**A**); and UV wavelengths of 366 nm (**B**).

**Figure 2 pharmaceuticals-16-00191-f002:**
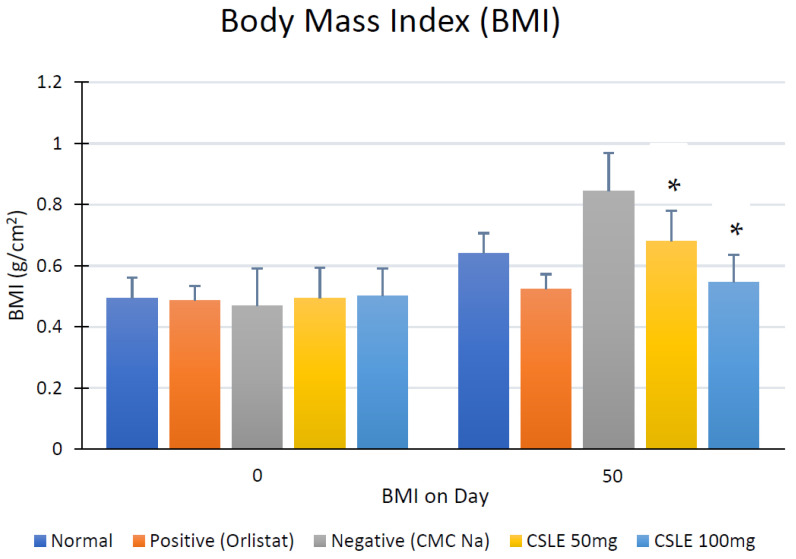
Animal body mass index (BMI) was compared between day 0 and day 50. * *p* < 0.05.

**Figure 3 pharmaceuticals-16-00191-f003:**
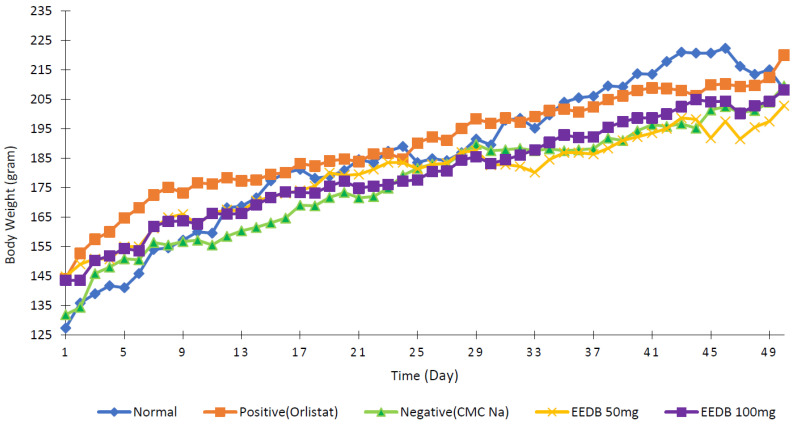
Rats’ body weight chart during the 50 days.

**Figure 4 pharmaceuticals-16-00191-f004:**
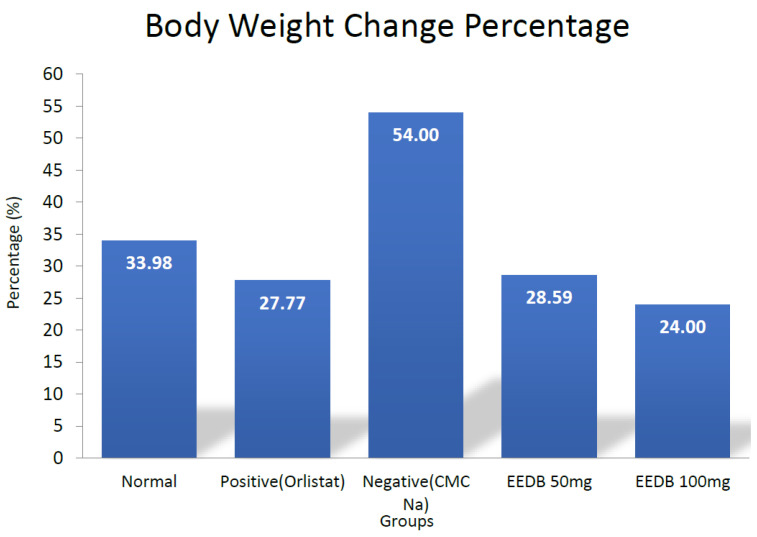
Body weight change percentage of each group after 50 days of HFD induction and 20 days of treatment compared to day 0.

**Figure 5 pharmaceuticals-16-00191-f005:**
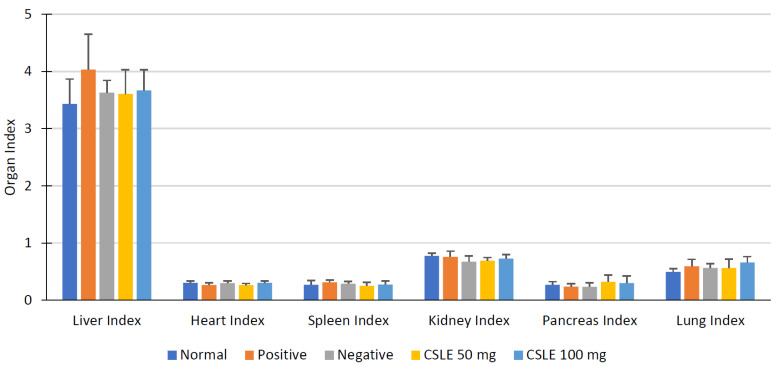
Organ index diagram.

**Figure 6 pharmaceuticals-16-00191-f006:**
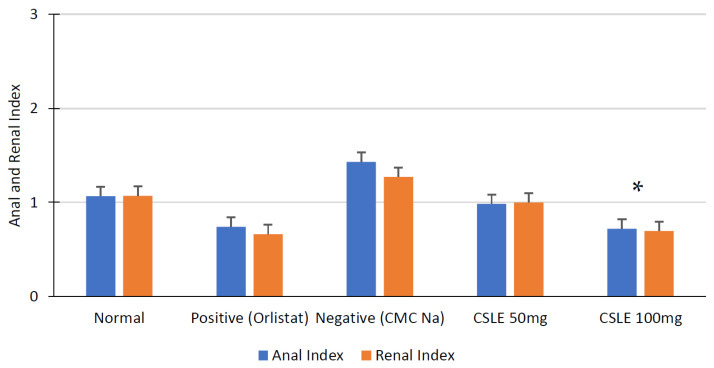
Carcass fat index diagram. The anal and renal fat index of CSLE dose of 100 mg/kgbw and orlistat group are not significantly different (* *p* < 0.05).

**Figure 7 pharmaceuticals-16-00191-f007:**
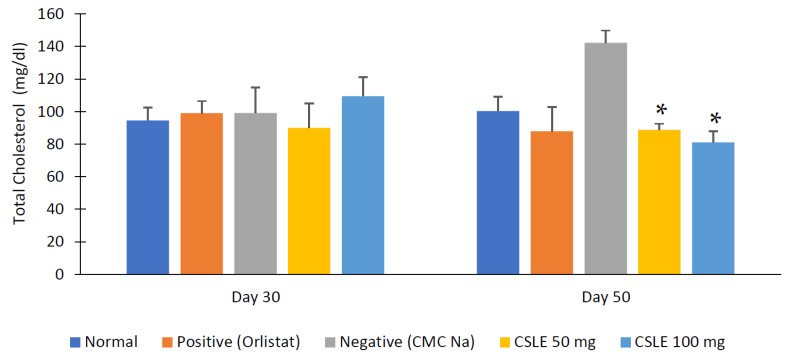
Total blood cholesterol diagram. Data represent mean ± SD (*n* = 5). The CSLE dose 50 and 100 mg/kgbw and orlistat group are not significantly different (* *p* < 0.05) by the ANOVA test.

**Figure 8 pharmaceuticals-16-00191-f008:**
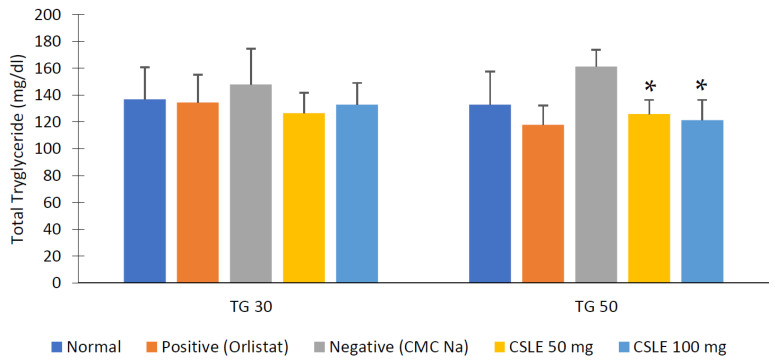
Total triglyceride value. Data represent mean ± SD (*n* = 5). CSLE dose 50 and 100 mg/kgbw and orlistat group are not significantly different (* *p* < 0.05) by the ANOVA test.

**Figure 9 pharmaceuticals-16-00191-f009:**
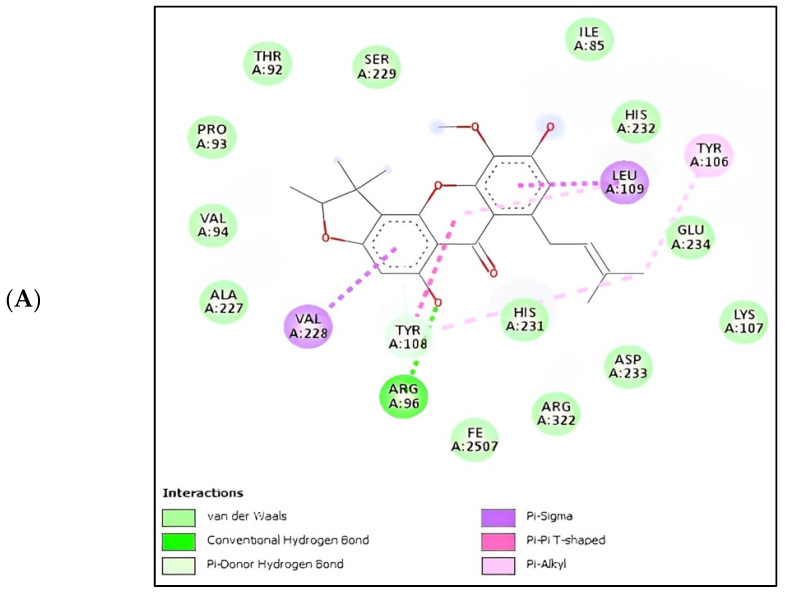
Docking visualisation of the interaction in 2D between Caloxanthone B (**A**) and Orlistat (**B**) with FTO by molecular docking simulation and residues of the binding site.

**Figure 10 pharmaceuticals-16-00191-f010:**
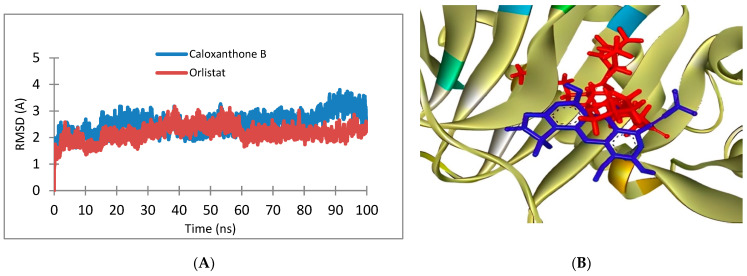
(**A**) Graphic of Root Mean Square Deviation (RMSD) of caloxanthone B and orlistat; (**B**) 3D interaction of caloxanthone B (Blue) and orlistat (Red) to fat mass and obesity-associated protein (FTO).

**Table 1 pharmaceuticals-16-00191-t001:** Specific and non-specific parameters of *Calophyllum soulattri* leaves extract.

No.	Parameter	Results
	Specific Parameter	
1.	Organoleptic	Thick, dark greenish, aromatic
2.	Water-soluble extract (%)	23.1
3.	Ethanol-soluble extract (%)	24.0
	Nonspecific parameters	
1.	Density (g/mL)	0.8033
2.	Drying losses (%)	16.32 ± 0.66
3.	Water content (%)	12.76

**Table 2 pharmaceuticals-16-00191-t002:** Phytochemical screening parameters of *Calophyllum soulattri* leaves extract.

No.	Parameter	Results *
1.	Alkaloid	+
2.	Polyphenol	+
3.	Tannin	−
4.	Flavonoid	+
5.	Steroid-triterpenoid	+
6.	Saponin	+

* plus (+) indicates the presence and minus (−) signifies absence.

**Table 3 pharmaceuticals-16-00191-t003:** Foaming and fish index of *Calophyllum soulattri* leaves.

No	Sample	Foaming Index	Fish Index
1.	Powdered CSL	<100	200
2.	CSLE	166.67	400

**Table 4 pharmaceuticals-16-00191-t004:** Drug-likeness prediction of the compounds in *Calophyllum soulattri*.

Selected Compound	Mr	XlogP3 AA	H Donor	H Akseptor	Chemical Structure
Friedelin	426.7	9.8	0	1	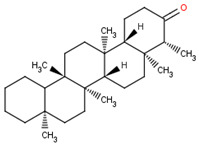
Caloxanthone B	410.5	6	2	6	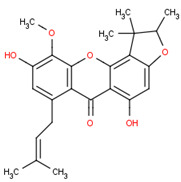
Macluraxanthone	394.4	5.3	3	6	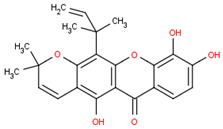
Stigmasterol	412.7	8.6	1	1	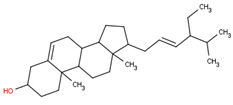
Trapezifolixanthone	378.4	5.7	2	5	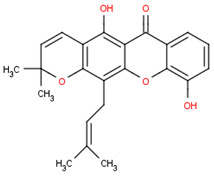
Dombakinaxanthone	446.5	7.6	2	5	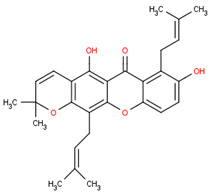
Brasixanthone B	378.4	5.7	2	5	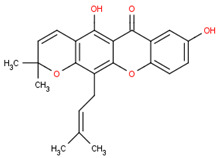
Orlistat	495.7	10	1	5	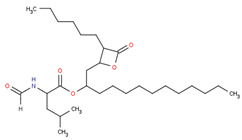

**Table 5 pharmaceuticals-16-00191-t005:** Result of molecular docking of selected compounds in *Calophyllum soulattri* to fat mass and obesity-associated protein (FTO).

Selected Compound	△*G* (Kcal/mol)
Caloxanthone B	–9.74
Brasixanthone B	–9.39
Stigmasterol	–9.34
Trapezifolixanthone	–8.85
Dombakinaxanthone	–8.68
Macluraxanthone	–8.48
Friedelin	–8.27
Ref. ligan	–6.53
Orlistat	–5.93

**Table 6 pharmaceuticals-16-00191-t006:** Predictive pharmacokinetic and toxicity parameters of selected compounds in *Calophyllum soulattri*.

Selected Compound	Water Solubility (log mol/L)	Intestinal Absorption (Human) (% Absorbed)	Distribution Volume (Human) (log L/kg)	Total Clearance (log mL/min/kg)	AMES Toxicity	Oral Rat Acute Toxicity (LD50) (mol/kg)	Hepatotoxicity
Friedelin	–5.52	98.74	–0.27	–0.04	No	2.64	No
Caloxanthone B	–5.00	94.82	–0.21	−0.035	Yes	1.87	No
Macluraxanthone	–3.58	91.98	0.2	0.08	Yes	2.02	No
Stigmasterol	–6.68	94.97	0.18	0.62	No	2.54	No
Trapezifolixanthone	–4.32	95.46	0.29	0.09	No	1.89	No
Dombakinaxanthone	–5.24	93.21	0.06	–0.11	Yes	1.79	Yes
Brasixanthone B	–4.41	95.31	0.49	0.11	Yes	1.95	No
Orlistat	–5.29	90.58	–0.55	1.68	No	1.97	Yes

**Table 7 pharmaceuticals-16-00191-t007:** Interaction energy of orlistat and caloxanthone B to fat mass and obesity-associated protein (FTO).

Component of Energy	Orlistat (kcal/mol)	Caloxanthone_B (kcal/mol)
van der Waals	–54.92	–34.22
Electrostatic	–15.15	–15.61
Electrostatic Poisson-Boltzmann	42.16	30.92
Nonpolar	–6.30	–3.63
Delta G Binding	–34.21	–22.54

## Data Availability

The data is contained within the article.

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
