# Peer review of "In Vivo Pharmacodynamics of Calophyllum soulattri as Antiobesity with In Silico Molecular Docking and ADME/Pharmacokinetic Prediction Studies"

_pharmaceuticals, 2023, doi:10.3390/ph16020191_

Round 1

Reviewer 1 Report

This manuscript is potentially interesting but there are some issues that should be carefully addressed by authors before making the paper suitable for publication in the Pharmaceuticals.

The English language requires revision by someone fluent in that language

In abstract, “group are 0.68, .57 and 0.52”. The 0 should be added before .57.

The introduction part is lacking about the description of plant. It can be improved by including some more data about the importance and literature review of the plant.

Have the authors investigated the toxicity of the extract.

Table 1 should be separated for the phytochemical screening. It should be partitioned in two tables.

The footnotes of the tables should be provided to make these tables self-explanatory.

There is no description in the results section. The description should be provided along with these tables.

Statistical data is also not provided for some data.

In figure 1, there is no labelling of A and B. Only in legend is describing A and B. This comment is also for figure 10.

There is included the methodology part in the discussion part. It should be omitted.

The line 92 and 93 require citation.

In graphs and figures, the description of Y-axis is bold in some places and at other places, it’s not bold. There should be uniformity.

From where the compounds were selected for molecular docking and pharmacokinetic prediction?

The resolution of figure 9 and 10 should be improved.

Why 96 % ethanol was used for extraction? Give the reference for this methodology.

Provide the DOI for the FTO and also provide the accession date for FTO and ligands from their data banks.

The authors have provided no reference for their any of study. They should include reference in each study protocol.

Author Response

Author'sNotes toReviewer1

Dear Reviewer 1,

We thank you for your time and effort in reviewing our manuscript. The feedback has been invaluable in improving the content and presentation of the paper.

We have revised our manuscript according to all of the academic reviewer or editor's comments.

  1. The English language requires revision by someone fluent in that language

Response:

The english grammar has been revised by an expert in english language. We used GoodLINGUAas our proofreader.

  1. In abstract, “group are 0.68, .57 and 0.52”. The 0 should be added before .57.

Response: This was revised to 0.57 on line 27.

  1. The introduction part is lacking about the description of plant. It can be improved by including some more data about the importance and literature review of the plant.

       Response :

In the introduction section, we improved the references to the Calophyllum soulattri description from line53to line59.The information related to its chemical compound and its in vivoactivity related to our study was added as well.

Calophyllum soulattriis an endemic plant in Indonesia, and its leaves extract is commonly used for treatment of obesity containing. It contains several secondary me-tabolite compounds, including flavonoid, steroid, triterpenoid, phenol, tannin, and saponin  [9] Friedelin, a pentacyclic terpenoid constituent, washave been determined in Calophyllum soulattrileaves extract, where, and this compound was assumedly involved in the the hypolipidemic activity of Azima tetracanthaLam. In rodent hyperlipidemia experimental models”

  1. Have the authors investigated the toxicity of the extract.

Response :

Yes, we have. We performed an acute and subchronic toxicity study of Calophyllum soulattri leaves extract (CSLE) in our previous study. These toxicity results have been added tointroductionsection from line64 to line70. 

“Acute and subchronic toxicity studies of Calophyllum soulattrileaves extract revealed no mortality or observed clinical signs of toxicity during investigation periods with LD50  >5000 mg/kgBW, but hepar necrosis was associated with long-term use of high-dose CSLE, such as 1000 mg/kgbw BWreference sucitransaminase (Fajriaty, et al., 2018). Moreover, the reported dose of Calophyllum soulattrileaves extract was between 100 and 400 mg/kgbwBW with no adverse effects were appeared on long-term use [12] (Fajriaty, et al., 2018).”

  1. Table 1 should be separated for the phytochemical screening. It should be partitioned in two tables.

Response :

We separated this table into two partitions. We put Table 1 as specific and non-specific parameters of Calophyllum soulattrileaves extract (CSLE) in line 98, and we put Table 2, phytochemical screening parameters for CSLE, in line 103.

Table 1. Specific and non-specific parameters of Calophyllum soulattrileaves extract

No.

Parameter

Results

Specific Parameter

1.

Organoleptic

Thick, dark greenish, aromatic

2.

Water-soluble extract (%)

23.1

3.

Ethanol-soluble extract (%)

24.0

Nonspecific parameters

1.

Density (g/ml)

0.8033

2.

Drying losses (%)

16.32 ± 0.66

3.

Water content (%)

12.76

Table 2. Phytochemical screening parameters of Calophyllum soulattrileaves extract

No.

Parameter

Results*

1.

Alkaloid

+

2.

Polyphenol

+

3.

Tannin

-

4.

Flavonoid

+

5.

Steroid-triterpenoid

+

6.

Saponin

+

*plus (+) indicates the presence and minus (-) signifies absence.

  1. The footnotes of the tables should be provided to make these tables self-explanatory.

Response : The Footnotes of Table 2 was provided.

The footnotes of Table 2 were provided.

  1. There is no description in the results section. The description should be provided along with these tables.

Response :

  1. Table 1 in Line 80-95

The specific and non-specific parameters results of the 96% ethanolic extract of Calophyllum soulattri leaves (CSLE) are shown in Table 1. We performed organoleptic determination of CSLE and its solubility towards two different organic solvents as specific parameter of extract. Organoleptic character describes colour, odour and texture, while the solubility parameters is to determine the availability of polar and semi-polar (or non-polar) compounds in the final extract. The results showed similar percentage value of the CSLE contents that were soluble in water and ethanol, such as 23.1% and 24.0%, respectively. Furthermore, using polar and semipolar solvents will result in a favourable extraction process.

Non-specific parameters of CSLE, including density extract, drying losses, and water content have been performed as well with their values are 0.8033 g/mL, 16.32 ± 0.66% and 12.76, respectively (Table 1). The Indonesian Herbal Pharmacopoeia specifies a minimum water content of 10% for extract, showing that the CSLE used in this study satisfies the requirements for standard quality [16] 

  1. Table 2 in Line 99-102

A review of the constituent compounds of CSLE was intended through phytochemical screening. According to phytochemical screening results, the 96% ethanolic extract of Calophyllum soulattri leaves contained alkaloids, polyphenols, flavonoids, steroid-triterpenoids, and saponins.

  1. Table 3 in Line 105-109

Mechanically agitated foam often results in an unsteady thermodynamic system. The foam set will degrade while it is inactive. Furthermore, tThe stability of thee foam is determined by its thinning rate [17][tambahkan referensi lenken17]. Table 3 displays the height change of the foam over time. CSLE's foaming index and fish index are higher compared to those of powdered CSL as a result.A review of the constituent compounds was intended through phytochemical screening. According to phytochemical screening results, the 96% ethanolic extract of Calophyllum soulattri leaves contained alkaloids, polyphenols, flavonoids, steroid-triterpenoids, and saponins.

  1. Lunkenheimer K, Malysa K. Simple and generally applicable method of determination and evaluation of foam properties. J. Surf. Deterg. 2003;6:69–74.
  2. Table 4 in Line 114-117

The drug-likeness parameter is an analysis to predict the physico-chemical properties of a drug based on its molecular weight (MW), Log P, the number of hydrogen donor (H donor) and acceptor (H acceptor). Based on these parameters, shown in Table 4, all seven compounds in Calophyllum soulattri and orlistat complied with Lipinski's rule. (MW <500, H donor <5, and H acceptor <10).

  1. Table 5 in Line 123-130

The molecular docking simulation results were determined by the Gibbs free energy value (ΔG), where the compound with the lowest ΔG value was predicted to have the strongest and most stable binding to the receptor. Molecular docking results showed that the seven compounds in Calophyllum soulattri have a better binding affinity than orlistat as a positive control, with the lowest ΔG are shown by cCaloxanthone B.  having the lowest ΔG value based on molecular docking. Moreover, this energy affinity describes the bond energy that occurs between the compounds in Calophyllum soulattri and the FTO receptor.

  1. Table 6 in Line 132-145

Table 6 explained the pharmacokinetics of selected compounds of Calophyllum soulattri, including their water solubility, intestinal absorption, volume of distribution, total clearance, and the prediction of their toxicity, including general toxicity (LD50) and specific toxicity (AMES toxicity test and hepatotoxicity). Overall, intestinal absorption value above 90% was shown by aAll selected compounds have an intestinal absorption value above 90%. Based on the AMES toxicity results, 4 compounds are are predicted to be mutagensic, such as cCaloxanthone B, mMacluraxanthone, dDombakinaxanthone, and bBrasixanthone B, but further in vivo tests are required. However, tThe prediction of hepatotoxicity test showed that most of of the compounds are not hepatotoxictoxic, except dDombakinaxanthone. The prediction of pharmacokinetic and toxicity parameters was is performed based on the similarity of the chemical structure of the predicted compounds to the chemical structure contained in the pkCSM-Biosig Lab. These predictions provide a basis for analysing pharmacokinetic and toxicity parameters based on seven potential compounds in Calophyllum soulattri.

  1. Statistical data is also not provided for some data.

Response : We did not use statistics tools to processin silicodata. But we provide statistical data for in vivoresults.

  1. In figure 1, there is no labelling of A and B. Only in legend is describing A and B. This comment is also for figure 10.

Response :

In Figure 1 in Line 160, Figure 9 in line 326and Figure 10 in Line331were labelled of A and B

  1. There is included the methodology part in the discussion part. It should be omitted.

Response : We revised the discussion part.

  1. The line 92 and 93 require citation.

Response :

In Line 404, citation were added with the reference below:

Lunkenheimer K, Malysa K. Simple and generally applicable method of determination and evaluation of foam properties. J. Surf. Deterg. 2003;6:69–74

  1. In graphs and figures, the description of Y-axis is bold in some places and at other places, it’s not bold. There should be uniformity.

Response :

It has been revised. It can be seen in line 332.

Figure 10A

  1. From where the compounds were selected for molecular docking and pharmacokinetic prediction?

Response :

It has been added in line 299-302.

These seven compounds were selected based on the previous study related to the phy-tochemical determination of Calophyllum soulattri that has been established and was provided in the Pubchem data bank [13–15,28,29].These study are mentioned in Introduction part in line 61-64.

  1. The resolution of figure 9 and 10 should be improved.

Response :

Figures 9 and 10 were improved.

Figure 9A Interaction 2D between Caloxanthone B and FTO by molecular docking simulation

Figure 9B Interaction 2D between Orlistat and FTO by molecular docking simulation

Figure 10B

  1. Why 96 % ethanol was used for extraction? Give thereference for this methodology.

Response :

We revised and added the referencesin line 374-380.

Redfern J, Kinninmonth M, Burdass D, Verran J. Using soxhlet ethanol extraction to produce and test plant material (essential oils) for their antimicrobial properties. J Microbiol Biol Educ. 2014 May 1;15(1):45-6. doi: 10.1128/jmbe.v15i1.656. PMID: 24839520; PMCID: PMC4004744.

  1. Provide the DOI for the FTO and also provide the accession date for FTO and ligands from their data banks.

Response :

We added the DOI and the accession date in the methodology sectionin Line 442-444:

DOI for the FTO PDB DOI: 10.2210/pdb3LFM/pdb accessed on November,12th2021

Ligands accessed on November, 3rd2021

  1. The authors have provided no reference for their any of study. They should include reference in each study protocol.

Response : We revised and added the references

All authors have read and approved the changes made to the manuscript. We hope that the revised paper is now suitable for inclusion in thePharmaceuticaland we look forward to hearing from you.

Yours sincerely,

Inarah Fajriaty

Reviewer 2 Report

1. Line 18-19 need revision

2. Lin 55; in vivo and in silico should change as

3. Table 1;Which are specific and which are nonspecific parameters?

4. What is meant by Alkaloid (+); polyphenol (+); tannin (-)???

5. Figure 1: Which one is A and B????

6. Graphical presentation should be uniform through out the manuscript

7. I think results section should be start with text not with table.

8.In Figure 2; either significant level of all results not calculated??

9. Figure 3 should be improved especially in term of Y-axis

10. There is need of improvement in Fig 10

11. Line 321-322; Confusion in the text also give the reference of the test

12. On which bases authors have selected these phytochemicals??? Explain

Author Response

Dear Reviewer 2,

We thank you for your time and effort in reviewing our manuscript. The feedback has been invaluable in improving the content and presentation of the paper.

We have revised our manuscript according to all of the academic reviewer or editor's comments.

  1. Line 18-19 need revision

Response :

In line 18-21 was revised

This study aims to determine the antiobesity activity of Calophyllum soulattrileaves extract (CSLE) on high fat diet-fed rats (HFD) and to predict the molecular docking and, molecular dynamic, pharmacokinetic and toxicity of  selected compounds of Calophyllum soulattrito fat mass and obesity-associated protein (FTO).

  1. Line 55; in vivo and in silico should change as

Response :

Line 71was changed in vivo and in silico.

  1. Table 1;Which are specific and which are nonspecific parameters?

Response :

We separated table 1 into two partitions as specific and non-specific parameters results of Calophyllum soulattrileaves extract (CSLE).

Table 1. Specific and non-specific parameters of Calophyllum soulattrileaves extract

No.

Parameter

Results

Specific Parameter

1.

Organoleptic

Thick, dark greenish, aromatic

2.

Water-soluble extract (%)

23.1

3.

Ethanol-soluble extract (%)

24.0

Nonspecific parameters

1.

Density (g/ml)

0.8033

2.

Drying losses (%)

16.32 ± 0.66

3.

Water content (%)

12.76

  1. What is meant by Alkaloid (+); polyphenol (+); tannin (-)???

Response :

That means“plus (+) indicates the presence and minus (-) signifies absence” and is providedin footnotes in Table 2 in line 108.

  1. Figure 1: Which one is A and B????

Response :

We added labels A and B in Figure 1, Figure 9, and Figure 10.

In Figure 1 in Line 160, Figure 9 in line 326and Figure 10 in Line331were labelled of A and B

  1. Graphical presentation should be uniform through out the manuscript

Response :

All graphics have been revised and uniformed.

  1. I think results section should be start with text not with table.

Response :

The results section has been revised and startedwith the text.

“The specific and non-specific parameters results of the 96% ethanolic extract of Calophyllum soulattri leaves (CSLE) are shown in Table 1. We performed organoleptic determination of CSLE and its solubility towards two different organic solvents as specific parameter of extract. Organoleptic character describes colour, odour and texture, while the solubility parameters is to determine the availability of polar and semi-polar (or non-polar) compounds in the final extract. The results showed similar percentage value of the CSLE contents that were soluble in water and ethanol, such as 23.1% and 24.0%, respectively. Furthermore, using polar and semipolar solvents will result in a favourable extraction process.

Non-specific parameters of CSLE, including density extract, drying losses, and water content have been performed as well with their values are 0.8033 g/mL, 16.32 ± 0.66% and 12.76, respectively (Table 1). The Indonesian Herbal Pharmacopoeia specifies a minimum water content of 10% for extract, showing that the CSLE used in this study satisfies the requirements for standard quality [16].

  1. In Figure 2; either significant level of all results not calculated??

Response :

In Figure 2, in line 186, "All significant levels of all results have been calculated."

  1. Figure 3 should be improved especially in term of Y-axis

Response :

Figure 3 has been improved, especially in terms of the Y-axis in Line 202.

  1. There is need of improvement in Fig 10

Response :

Figure 10 in Line 331was improved and labelled with A and B.

  1. Line 321-322; Confusion in the text also give the reference of the test

Response :

We revised the text and added the references for this sectionin line 410:

Apandi, M. (1984). Teknologi buah dan sayur / Muchidin Apandi. Bandung: Alumni.

Fajriaty I, IH Harianto, Andres, dan Setyaningrum R. Skrining Fitokimia dan Analisis Kromatografi Lapis dari Ekstrak Etanol Daun Bintangur (Calophyllum soulattri Burm.F.). Jurnal Pendidikan Informatika dan Sains. 2018; 7(1): 54-67.

  1. On which bases authors have selected these phytochemicals??? Explain

Response :

It has been added in line 299-302.

These seven compounds were selected based on the previous study related to the phy-tochemical determination of Calophyllum soulattri that has been established and was provided in the Pubchem data bank [13–15,28,29].These study are mentioned in Introduction part in line 61-64.

All authors have read and approved the changes made to the manuscript. We hope that the revised paper is now suitable for inclusion in thePharmaceuticaland we look forward to hearing from you.

Yours sincerely,

Inarah Fajriaty

Reviewer 3 Report

1. The study is evaluating the anti-obesity effect of the plant Calophyllum soulattri, the introduction about the plant should be enriched. The introduction should cover important points like the plant belongs to which family, studies reporting the phytochemicals isolated from it, biological activities reported before....etc. It is highly advised that the introduction should be improved to detail those aspects.

2. Results didn't show why compounds in table 3 were selected, if they were identified or isolated previously from the plant, and are they the major components in the extract that's why you may refer that the anti-obesity effect of the extract is attributed to those results from the drug likeness prediction.

Author Response

Dear Reviewer 3,

We thank you for your time and effort in reviewing our manuscript. The feedback has been invaluable in improving the content and presentation of the paper.

We have revised our manuscript according to all of the academic reviewer or editor's comments.

  1. The study is evaluating the anti-obesity effect of the plant Calophyllum soulattri, the introduction about the plant should be enriched. The introduction should cover important points like the plant belongs to which family, studies reporting the phytochemicals isolated from it, biological activities reported before....etc. It is highly advised that the introduction should be improved to detail those aspects.

Response :

  1. In the introduction section, we improved the references to the Calophyllum soulattri description from line53to lineThe information related to its chemical compound and its in vivoactivity related to our study was added as well.

Calophyllum soulattriis an endemic plant in Indonesia, and its leaves extract is commonly used for treatment of obesity containing. It contains several secondary me-tabolite compounds, including flavonoid, steroid, triterpenoid, phenol, tannin, and saponin[9] Friedelin, a pentacyclic terpenoid constituent, washave been determined in Calophyllum soulattrileaves extract, where, and this compound was assumedly involved in the the hypolipidemic activity of Azima tetracanthaLam. In rodent hyperlipidemia experimental models”

  1. We performed an acute and subchronic toxicity study of Calophyllum soulattri leaves extract (CSLE) in our previous study. These toxicity results have been added tointroductionsection from line64 to line

“Acute and subchronic toxicity studies of Calophyllum soulattrileaves extract revealed no mortality or observed clinical signs of toxicity during investigation periods with LD50  >5000 mg/kgBW, but hepar necrosis was associated with long-term use of high-dose CSLE, such as 1000 mg/kgbw BWreference sucitransaminase (Fajriaty, et al., 2018). Moreover, the reported dose of Calophyllum soulattrileaves extract was between 100 and 400 mg/kgbwBW with no adverse effects were appeared on long-term use [12] (Fajriaty, et al., 2018).”

  1. Results didn't show why compounds in table 3 were selected, if they were identified or isolated previously from the plant, and are they the major components in the extract that's why you may refer that the anti-obesity effect of the extract is attributed to those results from the drug likeness prediction.

Response :

It has been added in line 299-302.

These seven compounds were selected based on the previous study related to the phy-tochemical determination of Calophyllum soulattri that has been established and was provided in the Pubchem data bank [13–15,28,29].These study are mentioned in Introduction part in line 61-64.

All authors have read and approved the changes made to the manuscript. We hope that the revised paper is now suitable for inclusion in thePharmaceuticaland we look forward to hearing from you.

Yours sincerely,

Inarah Fajriaty

Round 2

Reviewer 1 Report

The authors have improved their manuscript to significant level. Now this MS looked improved. However. the authors have not described the used abbreviations which have not described in the whole MS. In figures 3 and 4, they have written EEDB. This is not explained anywhere. All the figures and tables should be self explanatory.